# Learning by Abstraction: The Neural State Machine

**Drew A. Hudson**
Stanford University
353 Serra Mall, Stanford, CA 94305
`dorarad@cs.stanford.edu`

**Christopher D. Manning**
Stanford University
353 Serra Mall, Stanford, CA 94305
`manning@cs.stanford.edu`

## Abstract

We introduce the Neural State Machine, seeking to bridge the gap between the neural and symbolic views of AI and integrate their complementary strengths for the task of visual reasoning. Given an image, we first predict a probabilistic graph that represents its underlying semantics and serves as a structured world model. Then, we perform sequential reasoning over the graph, iteratively traversing its nodes to answer a given question or draw a new inference. In contrast to most neural architectures that are designed to closely interact with the raw sensory data, our model operates instead in an abstract latent space, by transforming both the visual and linguistic modalities into semantic concept-based representations, thereby achieving enhanced transparency and modularity. We evaluate our model on VQA-CP and GQA, two recent VQA datasets that involve compositionality, multi-step inference and diverse reasoning skills, achieving state-of-the-art results in both cases. We provide further experiments that illustrate the model's strong generalization capacity across multiple dimensions, including novel compositions of concepts, changes in the answer distribution, and unseen linguistic structures, demonstrating the qualities and efficacy of our approach.

## 1 Introduction

Language is one of the most marvelous feats of the human mind. The emergence of a compositional system of symbols that can distill and convey from rich sensory experiences to creative new ideas has been a major turning point in the evolution of intelligence, and made a profound impact on the nature of human cognition [19, 79, 13]. According to Jerry Fodor's Language of Thought hypothesis [22, 73], thinking itself posses a language-like compositional structure, where elementary concepts combine in systematic ways to create compound new ideas or thoughts, allowing us to make "infinite use of finite means" [18] and fostering human's remarkable capacities of abstraction and generalization [51].

Indeed, humans are particularly adept at making abstractions of various kinds: We make analogies and form **concepts** to generalize from given instances to unseen examples [71]; we see things in context, and build compositional **world models** to represent objects and understand their interactions and subtle relations, turning raw sensory signals into high-level semantic knowledge [65]; and we deductively draw inferences via conceptual rules and statements to proceed from known facts to novel conclusions [32, 40]. Not only are humans capable of learning, but we are also talented at **reasoning**.

Ideas about compositionality, abstraction and reasoning greatly inspired the classical views of artificial intelligence [75, 66], but have lately been overshadowed by the astounding success of deep learning over a wide spectrum of real-world tasks [33, 64, 83]. Yet, even though neural networks are undoubtedly powerful, flexible and robust, recent work has repeatedly demonstrated their flaws, showing how they struggle to generalize in a systematic manner [50], overly adhere to superficial and potentially misleading statistical associations instead of learning true causal relations [1, 42], strongly depend on large amounts of data and supervision [25, 51], and sometimes behave in surprising and

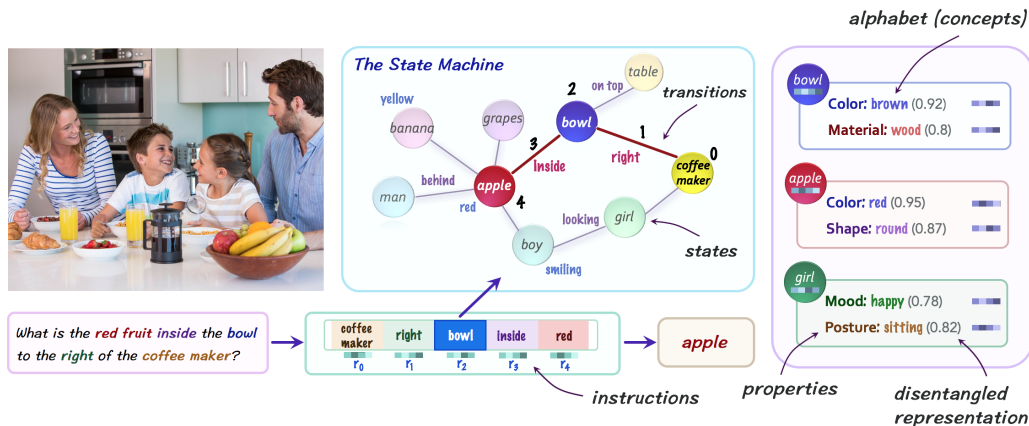

Figure 1: The Neural State Machine is a graph network that simulates the computation of an automaton. For the task of VQA, the model constructs a probabilistic scene graph to capture the semantics of a given image, which it then treats as a state machine, traversing its states as guided by the question to perform sequential reasoning.

worrisome ways [26, 20]. The sheer size and statistical nature of these models that support robustness and versatility are also what hinder their interpretability, modularity, and soundness.

Motivated to alleviate these deficiencies and bring the neural and symbolic approaches more closely together, we propose the Neural State Machine, a differentiable graph-based model that simulates the operation of an automaton, and explore it in the domain of visual reasoning and compositional question answering. Essentially, we proceed through two stages: *modeling* and *inference*. Starting from an image, we first generate a probabilistic scene graph [43, 49] that captures its underlying semantic knowledge in a compact form. Nodes correspond to objects and consist of structured representations of their properties, and edges depict both their spatial and semantic relations. Once we have the graph, we then treat it as a *state machine* and simulate an iterative computation over it, aiming to answer questions or draw inferences. We translate a given natural language question into a series of soft instructions, and feed them one-at-a-time into the machine to perform sequential reasoning, using attention to traverse its states and compute the answer.

Drawing inspiration from Bengio's consciousness prior [12], we further define a set of semantic embedded *concepts* that describe different entities and aspects of the domain, such as various kinds of objects, attributes and relations. These concepts are used as the vocabulary that underlies both the scene graphs derived from the image as well as the reasoning instructions obtained from the question, effectively allowing both modalities to "speak the same language". Whereas neural networks typically interact directly with raw observations and dense visual features, our approach encourages the model to reason instead in a semantic and factorized abstract space, which enables the disentanglement of structure from content and improves its modularity.

We demonstrate the value and performance of the Neural State Machine on two recent Visual Question Answering (VQA) datasets: GQA [41] which focuses on real-world visual reasoning and multi-step question answering, as well as VQA-CP [3], a recent split of the popular VQA dataset [2, 27] that has been designed particularly to evaluate generalization. We achieve state-of-the-art results on both tasks under single-model settings, substantiating the robustness and efficiency of our approach in answering challenging compositional questions. We then construct new splits leveraging the associated structured representations provided by GQA and conduct further experiments that provide significant evidence for the model's strong generalization skills across multiple dimensions, such as novel compositions of concepts and unseen linguistic structures, validating its versatility under changing conditions.

Our model ties together two important qualities: abstraction and compositionality, with the respective key innovations of representing meaning as a structured attention distribution over an internal vocabulary of disentangled concepts, and capturing sequential reasoning as the iterative computation of a differentiable state machine over a semantic graph. We hope that creating such neural form of a classical model of computation will encourage and support the integration of the connectionist and symbolic methodologies in AI, opening the door to enhanced modularity, versatility, and generalization.

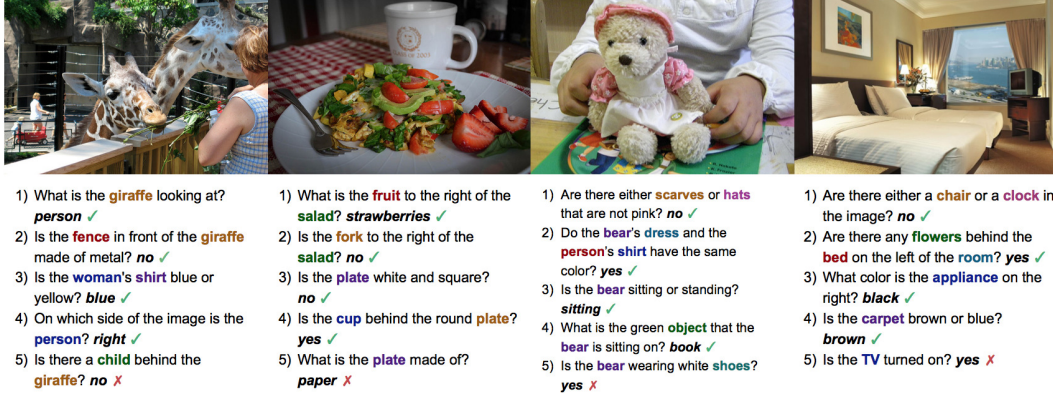

Figure 2: Question examples along with answers predicted by the NSM. The questions involve diverse reasoning skills such as multi-step inference, relational and spatial reasoning, logic and comparisons.

## 2 Related work

Our model connects to multiple lines of research, including works about compositionality [14, 38], concept acquisition [36, 82], and neural computation [28, 63, 7], which have explored the incorporation of structural priors into neural networks to promote interpretability and generalization. Recent research about scene graphs [43, 85] and graph networks [10] is also relevant to our work, where we propose a novel method for neural graph traversal that is more suitable than prior approaches to our goal of performing sequential reasoning, as it eliminates the need in this case for costly state updates, as in [56, 48, 77].

We explore our model in the context of VQA, a challenging multimodal task that has gained substantial attention in recent years [27, 80, 40]. Prior work commonly relied on dense visual features produced by either CNNs [84, 87] or object detectors [5], with a few recent models that use the relationships among objects to augment those features with contextual information from each object's surroundings [53, 76, 67]. We move further in this direction, performing iterative reasoning over inferred scene graphs, and in contrast to prior models, incorporate higher-level semantic concepts to represent both the visual and linguistic modalities in a shared and sparser manner that facilitates their interaction. For further discussion of related work please refer to the supplementary material, where we provide greater detail and additional information.

## 3 The Neural State Machine

The Neural State Machine is a graph-based network that simulates the computation of a finite automaton [37], and is explored here in the context of VQA, where we are given an image and a question and asked to provide an answer. We go through two stages – modeling and inference, the first to construct the state machine, and the second to simulate its operation.

In the modeling stage, we transform both the visual and linguistic modalities into abstract representations. The image is decomposed into a probabilistic *graph* that represents its semantics – the objects, attributes and relations in the depicted visual scene (section 3.2), while the question is converted into a sequence of reasoning *instructions* (section 3.3) that have to be performed in order to answer it.

In the inference stage (section 3.4), we treat the graph as a state machine, where the nodes, the objects within the image, correspond to states, and the edges, the relations between the objects, correspond to transitions. We then simulate a serial computation by iteratively feeding the machine with the instructions derived from the question and traversing its states, which allows us to perform sequential reasoning over the semantic visual scene, as guided by the question, to arrive at the answer.

We begin with a formal definition of the machine. In simple terms, a state machine is a computational model that consists of a collection of states, which it iteratively traverses while reading a sequence of inputs, as determined by a transition function. In contrast to the classical deterministic versions, the neural state machine defines an initial distribution over the states, and then performs a fixed number

of computation steps $N$, recurrently updating the state distribution until completion. Formally, we define the neural state machine as a tuple $(C, S, E, \{r_i\}_{i=0}^N, p_0, \delta)$:

- $C$ the model's alphabet, consisting of a set of concepts, embedded as learned vectors.
- $S$ a collection of states.
- $E$ a collection of directed edges that specify valid transitions between the states.
- $r_i$ a sequence of instructions, each of dimension $d$, that are passed in turn as an input to the transition function $\delta$.
- $p_0 : S \rightarrow [0, 1]$ a probability distribution of the initial state.
- $\delta_{S,E} : p_i \times r_i \rightarrow p_{i+1}$ a state transition function: a neural module that at each step $i$ considers the distribution $p_i$ over the states as well as an input instruction $r_i$, and uses it to redistribute the probability along the edges, yielding an updated state distribution $p_{i+1}$.

## 3.1 Concept vocabulary

In contrast to many common networks, the neural state machine operates over a discrete set of concepts. We create an embedded concept vocabulary $C$ for the machine (initialized with GloVe [69]), that will be used to capture and represent the semantic content of input images. The vocabulary is grouped into $L + 2$ *properties* such as object identity $C_O = C_0$ (e.g. *cat*, *shirt*), different types of attributes $C_A = \bigcup_{i=1}^L C_i$ (e.g. *colors*, *materials*) and relations $C_R = C_{L+1}$ (e.g. *holding*, *behind*), all derived from the Visual Genome dataset [49] (see section 6.3 for details). We similarly define a set of embeddings $D$ for each of the property types (such as *"color"* or *"shape"*).

In using the notion of concepts, we draw a lot of inspiration from humans, who are known for their ability to learn concepts and use them for tasks that involve abstract thinking and reasoning [11, 9, 30, 68]. In the following sections, rather than using raw and dense sensory input features directly, we represent both the visual and linguistic inputs in terms of our vocabulary, finding the most relevant concepts that they relate to. By associating such semantic concepts with raw sensory information from both the image and the question, we are able to derive higher-level representations that abstract away from irrelevant raw fine-grained statistics tied to each modality, and instead capture only the semantic knowledge necessary for the task. That way we can effectively cast both modalities onto the same space to facilitate their interaction, and, as discussed in section 4, improve the model's compositionality, robustness and generalization skills.

## 3.2 States and edge transitions

In order to create the state machine, we construct a probabilistic scene graph that specifies the objects and relations in a given image, and serves us as the machine's state graph, where objects correspond to states and relations to valid transitions. Multiple models have been proposed for the task of *scene graph generation* [85, 86, 16, 89]. Here, we largely follow the approaches of Yang et al. [86] and Chen et al. [16] in conjunction with a variant of the Mask R-CNN object detector [34] proposed by Hu et al. [39]. Further details regarding the graph generation can be found in section 6.4.

By using such a graph generation model, we can infer a scene graph that consists of: (1) A set of **object nodes** $S$ from the image, each accompanied by a bounding box, a mask, dense visual features, and a collection of discrete probability distributions $\{P_i\}_{i=0}^L$ for each of the object's $L + 1$ **semantic properties** (such as its color, material, shape, etc.), defined over the concept vocabulary $\{C_i\}_{i=0}^L$ presented above; (2) A set of **relation edges** between the objects, each associated with a probability distribution $P_{L+1}$ of its semantic type (e.g. *on top of*, *eating*) among the concepts in $C_{L+1}$, and corresponding to a valid transition between the machine's states.

Once we obtain the sets of state nodes and transition edges, we proceed to computing structured embedded representations for each of them. For each state $s \in S$ that corresponds to an object in the scene, we define a set of $L + 1$ property variables $\{s^j\}_{j=0}^L$ and assign each of them with

$$s^j = \sum_{c_k \in C_j} P_j(k)c_k$$

Where $c_k \in C_j$ denotes each embedded concept of the $j^{th}$ property type and $P_j$ refers to the corresponding property distribution over these concepts, resulting in a soft-binding of concepts to

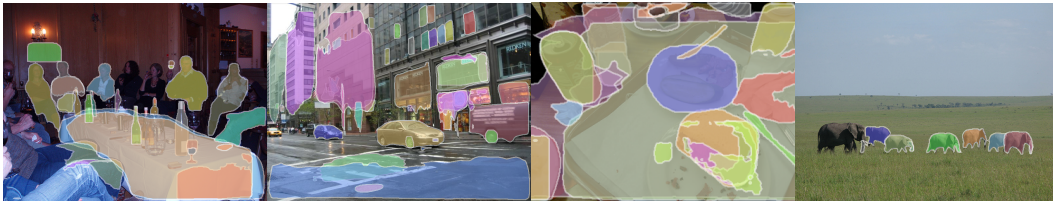

Figure 3: A visualization of object masks from the inferred scene graphs, which form the basis for our model.

each variable. To give an example, if an object is recognized by the object detector as likely to be e.g. *red*, then its *color* variable will be assigned to an averaged vector close to the embedding of the *"red"* concept. Edge representations are computed in a similar manner, resulting in matching embeddings of their relation type: $e' = \sum_{c_k \in C_{L+1}} P_{L+1}(k)c_k$ for each edge $e \in E$.

Consequently, we obtain a set of structured representations for both the nodes and the edges that underlie the state machine. Note that by associating each object and relation in the scene with not one, but a collection of vectors that capture each of their semantic properties, we are able to create disentangled representations that encapsulate the statistical particularities of the raw image and express it instead through a factorized discrete distribution over a vocabulary of embedded semantic concepts, aiming to encourage and promote higher compositionality.

### 3.3 Reasoning instructions

In the next step, we translate the question into a sequence of reasoning instructions (each expressed in terms of the concept vocabulary $C$), which will later be read by the state machine to guide its computation. The translation process consists of two steps: tagging and decoding.

We begin by embedding all the question words using GloVe (dimension $d = 300$). We process each word with a soft tagger function that either translates it into the most relevant concept in our vocabulary or alternatively keeps it intact, if it does not match any of them closely enough. Formally, for each embedded word $w_i$ we compute a similarity-based distribution

$$P_i = \text{softmax}(w_i^T \mathbf{W} C)$$

Where $\mathbf{W}$ is initialized to the identity matrix and $C$ denotes the matrix of all embedded concepts along with an additional learned default embedding $c'$ to account for structural or other non-content words.

Next, we translate each word into a concept-based representation:

$$v_i = P_i(c')w_i + \sum_{c \in C \setminus \{c'\}} P_i(c)c$$

Intuitively, a content word such as *apples* will be considered mostly similar to the concept *apple* (by comparing their GloVe embeddings), and thus will be replaced by the embedding of that term, whereas function words such as *who, are, how* will be deemed less similar to the semantic concepts and hence will stay close to their original embedding. Overall, this process allows us to normalize, or contextualize, the question, by transforming content words to their matching concepts, while keeping function words mostly unaffected.

Finally, we process the normalized question words with an attention-based encoder-decoder, drawing inspiration from [40]: Given a question of $M$ normalized words $V^{M \times d} = \{v_i\}_{i=1}^{M}$, we first pass it through an LSTM encoder, obtaining the final state $q$ to represent the question. Then, we roll-out a recurrent decoder for a fixed number of steps $N + 1$, yielding $N + 1$ hidden states $\{h_{i=0}^{N}\}$, and transform each of them into a corresponding reasoning instruction:

$$r_i = \text{softmax}(h_i V^T)V$$

Here, we compute attention over the normalized question words at each decoding step. By repeating this process for all $N + 1$ steps, we decompose the question into a series of reasoning instructions that selectively focus on its various parts, accomplishing the goal of this stage.

### 3.4 Model simulation

Having all the building blocks of the state machine ready, the graph of states $S$ and edges $E$, the instruction series $\{r_i\}_{i=0}^{N}$, and the concept vocabulary $C = \bigcup_{i=0}^{L+1} C_i$, we can now simulate the machine's sequential computation. Basically, we will begin with a uniform initial distribution $p_0$ over the states (the objects in the image's scene), and at each reasoning step $i$, read an instruction $r_i$ as derived from the question, and use it to redistribute our attention over the states (the objects) by shifting probability along the edges (their relations).

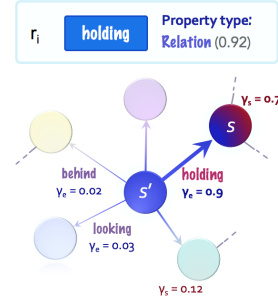

Formally, we perform this process by implementing a neural module for the state transition function $\delta_{S,E} : p_i \times r_i \to p_{i+1}$. At each step $i$, the module takes a distribution $p_i$ over the states as an input and computes an updated distribution $p_{i+1}$, guided by the instruction $r_i$. Our goal is to determine what next states to traverse to ($p_{i+1}$) based on the states we are currently attending to ($p_i$). To achieve that, we perform a couple of steps.

Figure 4: A visualization of a graph traversal step, where attention is being shifted from one node to its neighbor along the most relevant edge.

Recall that in section 3.2 we define for each object a set of $L + 1$ variables, representing its different properties (e.g. identity, color, shape). We further assigned each edge with a variable that similarly represents its relation type. Our first goal is thus to find the instruction *type*: the property type that is most relevant to the instruction $r_i$ – basically, to figure out what the instruction is about. We compute the distribution $R_i = \text{softmax}(r_i^T \circ D)$ over the $L + 2$ embedded properties $D$, defined in section 3.1. We further denote $R_i(L + 1) \in [0, 1]$ that corresponds to the relation property as $r_i'$, measuring the degree to which that reasoning instruction is concerned with semantic relations (in contrast to other possibilities such as e.g. objects or attributes).

Once we know what the instruction $r_i$ is looking for, we can use it as a guiding signal while traversing the graph from the current states we are focusing on to their most relevant neighbors. We compare the instruction to all the states $s \in S$ and edges $e \in E$, computing for each of them a relevance score:

$$\gamma_i(s) = \sigma\Big( \sum_{j=0}^{L} R_i(j)(r_i \circ \mathbf{W}_j s^j) \Big) \tag{1}$$

$$\gamma_i(e) = \sigma\big( r_i \circ \mathbf{W}_{L+1} e' \big) \tag{2}$$

Where $\sigma$ is a non-linearity, $\{s^j\}_{j=0}^{L}$ are the state variables corresponding to each of its properties, and $e'$ is the edge variable representing its type. We then get relevance scores between the instruction $r_i$ and each of the variables, which are finally averaged for each state and edge using $R_i$.

Having a relevance score for both the nodes and the edges, we can use them to achieve the key goal of this section: shifting the model's attention $p_i$ from the current nodes (states) $s \in S$ to their most relevant neighbors – the next states:

$$p_{i+1}^s = \text{softmax}_{s \in S}(\mathbf{W}_s \cdot \gamma_i(s)) \tag{3}$$

$$p_{i+1}^r = \text{softmax}_{s \in S}\Big(\mathbf{W}_r \cdot \sum_{(s',s) \in E} p_i(s') \cdot \gamma_i((s', s))\Big) \tag{4}$$

$$p_{i+1} = r_i' \cdot p_{i+1}^r + (1 - r_i') \cdot p_{i+1}^s \tag{5}$$

Here, we compute the distribution over the next states $p_{i+1}$ by averaging two probabilities $p_{i+1}^s$ and $p_{i+1}^r$: the former is based on each potential next state's own *internal* properties, while the latter considers the next states *contextual* relevance, relative to the current states the model attends to. Overall, by repeating this process over $N$ steps, we can simulate the iterative computation of the neural state machine.

After completing the final computation step, and in order to predict an answer, we use a standard 2-layer fully-connected softmax classifier that receives the concatenation of the question vector $q$ as well as an additional vector $m$ that aggregates information from the machine's final states:

$$m = \sum_{s \in S} p_N(s) \Big( \sum_{j=0}^{L} R_N(j) \cdot s^j \Big) \tag{6}$$

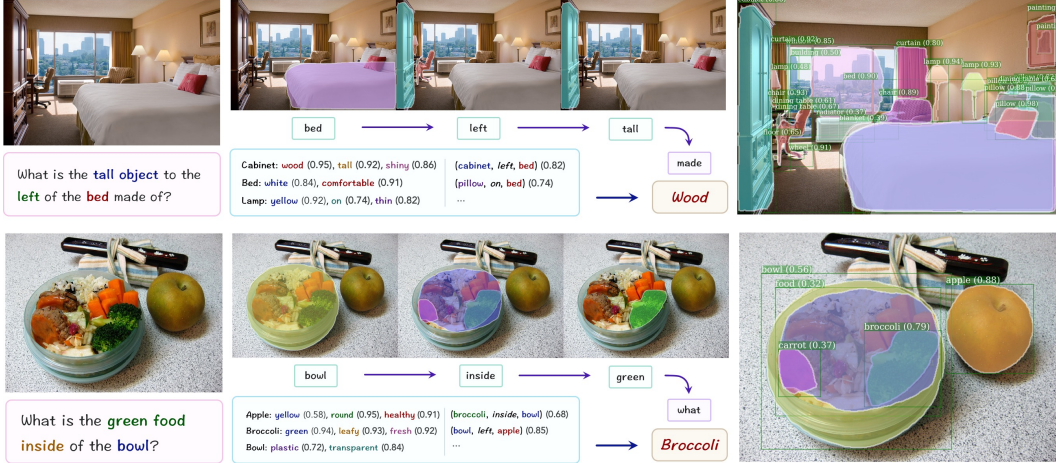

Figure 5: A visualization of the NSM's reasoning process: given an image and a question (left side), the model first builds a probabilistic scene graph (the blue box and the image on the right), and translates the question into a series of instructions (the green and purple boxes, where for each instruction we present its closest concept (or word) in vector space (section 3.1)). The model then performs sequential reasoning over the graph, attending to relevant object nodes in the image's scene as guided by the instructions, to iteratively compute the answer.

Where $m$ reflects the information extracted from the final states as guided by the final reasoning instruction $r_N$: averaged first by the reasoning instruction *type*, and then by the attention over the final states, as specified by $p_N$.

Overall, the above process allows us to perform a differentiable traversal over the scene graph, guided by the sequence of instructions that were derived from the question: Given an image and a question, we have first inferred a graph to represent the objects and relations in the image's scene, and analogously decomposed the question into a sequence of reasoning instructions. Notably, we have expressed both the graph and the instructions in terms of the shared vocabulary of semantic concepts, translating them both into the same "internal language". Then, we simulate the state machine's iterative operation, and over its course of computation, are successively shifting our attention across the nodes and edges as we ground each instruction in the graph to guide our traversal. Essentially, this allows us to locate each part of the question in the image, and perform sequential reasoning over the objects and relations in the image's scene graph until we finally arrive at the answer.

## 4 Experiments

We evaluate our model (NSM) on two recent VQA datasets: (1) The GQA dataset [41] which focuses on real-world visual reasoning and compositional question answering, and (2) VQA-CP (version 2) [3], a split of the VQA dataset [27] that has been particularly designed to test generalization skills across changes in the answer distribution between the training and the test sets. We achieve state-of-the-art performance both for VQA-CP, and, under single-model settings, for GQA. To further explore the generalization capacity of the NSM model, we construct two new splits for GQA that test generalization over both the questions' content and structure, and perform experiments based on them that provide substantial evidence for the strong generalization skills of our model across multiple dimensions. Finally, performance diagnosis, ablation studies and visualizations are presented in section 6.2 to shed more light on the inner workings of the model and its qualitative behavior.

Both our model and implemented baselines are trained to minimize the cross-entropy loss of the predicted candidate answer (out of the top 2000 possibilities), using a hidden state size of $d = 300$ and, unless otherwise stated, length of $N = 8$ computation steps for the MAC and NSM models. Please refer to section 6.5 for further information about the training procedure, implementation details, hyperparameter configuration and data preprocessing, along with complexity analysis of the NSM model. The model has been implemented in Tensorflow, and will be released along with the features and instructions for reproducing the described experiments.

Table 1: GQA scores for the single-model settings, including official baselines and top submissions

| Model | Binary | Open | Consistency | Validity | Plausibility | Distribution | Accuracy |
|---|---|---|---|---|---|---|---|
| Human [41] | 91.20 | 87.40 | 98.40 | 98.90 | 97.20 | - | 89.30 |
| Global Prior [41] | 42.94 | 16.62 | 51.69 | 88.86 | 74.81 | 93.08 | 28.90 |
| Local Prior [41] | 47.90 | 16.66 | 54.04 | 84.33 | 84.31 | 13.98 | 31.24 |
| Language [41] | 61.90 | 22.69 | 68.68 | 96.39 | **87.30** | 17.93 | 41.07 |
| Vision [41] | 36.05 | 1.74 | 62.40 | 35.78 | 34.84 | 19.99 | 17.82 |
| Lang+Vis [41] | 63.26 | 31.80 | 74.57 | 96.02 | 84.25 | 7.46 | 46.55 |
| BottomUp [5] | 66.64 | 34.83 | 78.71 | 96.18 | 84.57 | 5.98 | 49.74 |
| MAC [40] | 71.23 | 38.91 | 81.59 | 96.16 | 84.48 | 5.34 | 54.06 |
| SK T-Brain* | 77.42 | 43.10 | 90.78 | 96.26 | 85.27 | 7.54 | 59.19 |
| PVR* | 77.69 | 43.01 | 90.35 | **96.45** | 84.53 | 5.80 | 59.27 |
| GRN | 77.53 | 43.35 | 88.63 | 96.18 | 84.71 | 6.06 | 59.37 |
| Dream | 77.84 | 43.72 | 91.71 | 96.38 | 85.48 | 8.40 | 59.72 |
| LXRT | 77.76 | 44.97 | 92.84 | 96.30 | 85.19 | 8.31 | 60.34 |
| **NSM** | **78.94** | **49.25** | **93.25** | **96.41** | 84.28 | **3.71** | **63.17** |

## 4.1 Compositional question answering

We begin by testing the model on the GQA task [41], a recent dataset that features challenging compositional questions that involve diverse reasoning skills in real-world settings, including spatial reasoning, relational reasoning, logic and comparisons. We compare our performance both with baselines, as appear in [41], as well as with the top-5 single and top-10 ensemble submissions to the GQA challenge.[1] For single-model settings, to have a fair comparison, we consider all models that, similarly to ours, did not use the strong program supervision as an additional signal for training, but rather learn directly from the questions and answers.

As table 1 shows, we achieve state-of-the-art performance for a single-model across the dataset's various metrics (defined in [41]) such as accuracy and consistency. For the ensemble setting, we compute a majority vote of 10 instances of our model, achieving the 3rd highest score compared to the 52 submissions that have participated in the challenge[1] (table 2), getting significantly stronger scores compared to the 4th or lower submissions.

Note that while several submissions (marked with *) use the associated functional programs that GQA provides with each question as a strong supervision during train time, we intentionally did not use them in training our model, but rather aimed to learn the task directly using the question-answer pairs only. These results serve as an indicator for the ability of the model to successfully address questions that involve different forms of reasoning (see section 6 for examples), and especially multi-step inference, which is particularly common in GQA.

## 4.2 Generalization experiments

Motivated to measure the generalization capacity of our model, we perform experiments over three different dimensions: (1) changes in the answer distribution between the training and the test sets, (2) contextual generalization for concepts learned in isolation, and (3) unseen grammatical structures.

First, we measure the performance on VQA-CP [3], which provides a new split of the VQA2 dataset [27], where the answer distribution is kept different between the training and the test sets (e.g. in the training set, the most common color answer is *white*, whereas in the test set, it is *black*). Such settings reduce the extent to which models can circumvent the need for genuine scene understanding skills by exploiting dataset biases and superficial statistics [1, 44, 27], and are known to be particularly difficult for neural networks [51]. Here, we follow the standard VQA1/2 [27] accuracy metric for this task (defined in [3]). Table 3 presents our performance compared to existing approaches. We can see that NSM surpasses alternative models by a large margin.

We perform further generalization studies on GQA, leveraging the fact that the dataset provides grounding annotations of the question words. For instance, a question such as *"What color is the book on the table?"* is accompanied by the annotation $\{4 : (\text{"book"}, n_0), 7 : (\text{"table"}, n_1)\}$ expressing the fact that e.g. the $4^{th}$ word refers to the *book* object node. These annotations allow us to split the

Table 2: GQA ensemble

| Model | Accuracy |
|---|---|
| Kakao* | **73.33** |
| 270 | 70.23 |
| NSM | 67.25 |
| LXRT | 62.71 |
| GRN | 61.22 |
| MSM | 61.09 |
| DREAM | 60.93 |
| SK T-Brain* | 60.87 |
| PKU | 60.79 |
| Musan | 59.93 |

Table 3: VQA-CPv2

| Model | Accuracy |
|---|---|
| SAN [87] | 24.96 |
| HAN [60] | 28.65 |
| GVQA [3] | 31.30 |
| RAMEN [74] | 39.21 |
| BAN [46] | 39.31 |
| MuRel [15] | 39.54 |
| ReGAT [52] | 40.42 |
| **NSM** | **45.80** |

Table 4: GQA generalization

| Model | Content | Structure |
|---|---|---|
| Global Prior | 8.51 | 14.64 |
| Local Prior | 12.14 | 18.21 |
| Vision | 17.51 | 18.68 |
| Language | 21.14 | 32.88 |
| Lang+Vis | 24.95 | 36.51 |
| BottomUp [5] | 29.72 | 41.83 |
| MAC [40] | 31.12 | 47.27 |
| **NSM** | **40.24** | **55.72** |

| Structure Generalization | | Content Generalization | |
|---|---|---|---|
| training | testing | training | testing |
| What is the <obj> covered by? | What is covering the <obj>? | Only questions that do not refer to any type of food or animal (do not include any word from these categories) | Only questions that refer to foods or animals (include a word from one of these categories) |
| Is there a <obj> in the image? | Do you see any <obj>s in the photo? | | |
| What is the <obj> made of? | What material makes up the <obj>? | | |
| What's the name of the <obj> that is <attr>? | What is the <attr> <obj> called? | | |

Figure 6: Our new generalization splits for GQA, evaluating generalization over (1) content: where test questions ask about novel concepts, and (2) structure: where test questions follow unseen linguistic patterns.

training set in two interesting ways to test generalization over both *content* and *structure* (see figure 6 for an illustration of each split):

**Content**: Since the annotations specify which objects each question refers to, and by using the GQA ontology, we can identify all the questions that are concerned with particular object types, e.g. foods, or animals. We use this observation to split the training set by excluding all question-answer pairs that refer to these categories, and measure the model's generalization over them. Note however, that the object detector module described in section 3.2 is still trained over all the scene graphs including those objects – rather, the goal of this split is to test whether the model can leverage the fact that it was trained to identify a particular object in isolation, in order to answer unseen questions about that type of object without any further question training.

**Structure**: We can use the annotations described above as masks over the objects (see figure 6 for examples), allowing us to divide the questions in the training set into linguistic pattern groups. Then, by splitting these groups into two separated sets, we can test whether a model is able to generalize from some linguistic structures to unseen ones.

Table 4 summarizes the results for both settings, comparing our model to the baselines released for GQA [41], all using the same training scheme and input features. We can see that here as well, NSM performs significantly better than the alternative approaches, testifying to its strong generalization capacity both over concepts it has not seen any questions about (but only learned in isolation), as well as over questions that involve novel linguistic structures. In our view, these results point to the strongest quality of our approach. several prior works have argued for the great potential of abstractions and compositionality in enhancing models of deep learning [8, 10]. Our results suggest that incorporating these notions may indeed be highly beneficial to creating models that are more capable in coping with changing conditions and can better generalize to novel situations.

# 5 Conclusion

In this paper, we have introduced the Neural State Machine, a graph-based network that simulates the operation of an automaton, and demonstrated its versatility, robustness and high generalization skills on the tasks of real-world visual reasoning and compositional question answering. By incorporating the concept of a state machine into neural networks, we are able to introduce a strong structural prior that enhances compositinality both in terms of the *representation*, by having a structured graph to serve as our world model, as well as in terms of the *computation*, by performing sequential reasoning over such graphs. We hope that our model will help in the effort to integrate symbolic and connectionist approaches more closely together, in order to elevate neural models from sensory and perceptual tasks, where they currently shine, into the domains of higher-level abstraction, knowledge representation, compositionality and reasoning.

## Footnotes

[1]The official leaderboard mixes up single-model and ensemble results – we present here separated scores for each track.

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
