[Supplementary Material]

# 6 Supplementary material

Figure 1: Accuracy of NSM and baselines for different structural and semantic question types of the GQA dataset. NSM surpasses the baselines for all question types, with the largest gains for **relational** and **comparative** questions, and high improvements for **attribute**, **logic** and **query** (open) questions.

## 6.1 Related work (full version)

Our model connects to multiple lines of research, including works about concept acquisition [36], visual abstractions [24, 82, 59, 58, 88], compositionality [14, 38], and neural computation [28, 63, 7]. Several works have explored the discovery and use of visual concepts in the contexts of reinforcement or unsupervised learning [35, 17] as well as in classical computer vision [21, 78]. Others have argued for the importance of incorporating strong inductive biases into neural architectures [14, 10, 6, 8], and indeed, there is a growing body of research that seeks to introduce different forms of structural priors inspired by computer architectures [29, 81, 40] or theory of computation [4, 23], aiming to bridge the gap between the symbolic and neural paradigms.

One such structural prior that underlies our model is that of the probabilistic scene graph [43, 49] which we construct and reason over to answer questions about presented images. Scene graphs provide a succinct representation of the image's semantics, and have been effectively used for variety of applications such as image retrieval, captioning or generation [43, 54, 45]. Recent years have witnessed an increasing interest both in scene graphs in particular [85, 86, 16, 89, 55] as well as in graph networks in general [10, 48, 77] – a family of graph-structured models in which information is iteratively propagated across the nodes to turn their representations increasingly more contextual with information from their neighborhoods. In our work, we also use the general framework of graph networks, but in contrast to common approaches, we avoid the computationally-heavy state updates per each node, keeping the graph states static once predicted, and instead perform a series of refinements of one global attention distribution over the nodes, resulting in a soft graph traversal which better suits our need for supporting sequential reasoning.

We explore our model in the context of visual question answering [31], a challenging multi-modal task that has gained substantial attention over the last years. Plenty of models have been proposed [5, 80, 40], focusing on visual reasoning or question answering in both abstract and real-world settings [27, 44, 72, 41]. Typically, most approaches use either CNNs [84, 87] or object detectors [5] to derive visual features which are then compared to a fixed-size question embedding obtained by an LSTM. A few newer models use the relationships among objects to augment features with contextual information from each object's surroundings [53, 76, 67]. We move further in this direction, performing iterative reasoning over the inferred scene graphs, and in contrast to prior models [52, 15], incorporate higher-level semantic concepts to represent both the visual and linguistic modalities in a shared and sparser manner to facilitate their interaction.

Other methods for visual reasoning such as [88, 61, 62] have similar motivation to ours of integrating neural and symbolic perspectives by learning and reasoning over structured representations. However, in contrast to our approach, those models heavily rely on symbolic execution, either through strong supervision of program annotations [62], non-differentiable python-based functions [88], or a

collection of hand-crafted modules specifically designed for each given task [61], and consequently, they have mostly been explored in artificial environments such as CLEVR [44]. Instead, our model offers a fully-neural and more general graph-based design that, as demonstrated by our experiments, successfully scales to real-world settings.

Closest to our work is a model called MAC [40], a recurrent network that applies attention-based operations to perform sequential reasoning. However, the Neural State Machine differs from MAC in two crucial respects: First, we reason over graph structures rather than directly over spatial maps of visual features, traversing the graph by successively shifting attention across its nodes and edges. Second, key to our model is the notion of semantic concepts that are used to express knowledge about both the visual and linguistic modalities, instead of working directly with the raw observational features. As our findings suggest, these ideas contribute significantly to the model's performance compared to MAC and enhance its compositionality, transparency and generalization skills.

## 6.2 Ablation studies

To gain further insight into the relative contributions of different aspects of our model to its overall performance, we have conducted multiple ablation experiments, as summarized in table 5 and figure 4. First, we measure the impact of using our new visual features (section 3.2) compared to the default features used by the official baselines [41]. Such settings result in an improvement of 1.27%, confirming that most of the gain achieved by the NSM model (with 62.95% overall) stems from its inherent architecture rather than from the input features.

Figure 2: Accuracy as a function of the number of computation steps. Performance is reported on the GQA Test-Dev split.

We then explore several ablated models: one where we do not perform any iterative simulation of the state machine, but rather directly predict the answer from its underlying graph structure (55.41%); and a second enhanced version where we perform a traversal across the states but without considering the typed relations among them, adopting instead a uniform fully-connected graph (58.83%). These experiments suggest that using the graph structure to represent the visual scene as well performing sequential reasoning over it by traversing its nodes, are both crucial to the model's overall accuracy and have positive impact on its performance.

Next, we evaluate a variant of the model where instead of using the concept-based representations defined in section 3.1, we fallback to the more standard dense features to represent the various graph elements, which results in an accuracy of 58.48%. Comparing this score to that of the default model's settings (62.95%) proves the significance of using the higher-level semantic representations to the overall accuracy of the NSM.

Finally, we measure the impact of varying the number of computation steps on the obtained results (figure 4), revealing a steady increase in accuracy as we perform a higher number of reasoning steps (until saturation for $N = 8$ steps). These experiments further validate the effectiveness of sequential computation in addressing challenging questions, and especially compositional questions which are at the core of GQA.

## 6.3 Concept vocabulary

We define a vocabulary $C$ of 1335 embedded concepts about object types $C_0$ (e.g. *cat*, *shirt*), attributes, grouped into $L$ types $\{C_i\}_{i=1}^L$ (e.g. *colors*, *materials*), and relations $C_{L+1}$ (e.g. *holding*, *on top of*), all initialized with GloVe [69]. Our concepts consist of 785 objects, 170 relations, and 303 attributes that are divided into 77 types, with most types being binary (e.g. *short/tall*, *light/dark*). All concepts are derived from the Visual Genome dataset [49], which provides human-annotated scene graphs for real-world images. We use the refined dataset supported by GQA [41], where the graphs are further cleaned-up and consolidated, resulting in a closed-vocabulary version of the dataset.

## 6.4 Scene graph generation

In order to generate scene graphs from given images, we re-implement a model that largely follows Yang et al. [86], Chen et al. [16] in conjunction with an object detector proposed by Hu et al. [39].

Table 1: Ablations (reported on GQA Test-Dev)

| Model | Accuracy |
|-------|----------|
| NSM (default) | $62.95 \pm 0.22$ |
| No concepts | $58.48 \pm 0.14$ |
| No relations (set) | $58.83 \pm 0.17$ |
| No traversal | $55.41 \pm 0.35$ |
| Visual features | $47.82 \pm 0.09$ |
| Baseline (Lang+Vis) | $46.55 \pm 0.13$ |

Specifically, we use a variant of Mask R-CNN [34, 39] to obtain object detections from each image, using Hu et al. [39]'s official implementation along with ResNet-101 [33] and FPN [57] for features and region proposals respectively, and keep up to $50$ detections, with a confidence threshold of $0.2$. We train the object detector (and the following graph generation model) over a cleaner version of Visual Genome scene graphs [49], offered as part of the GQA dataset [41]. In particular, the detector heads are trained to classify both the object class and category (akin to YOLO's hierarchical softmax [70]), as well as the object's attributes per each *property* type, resulting in a set of probability distributions $\{P_i\}_{i=0}^{L}$ over the object's various properties (e.g. its *identity*, *color* or *shape*).

Once we obtain the graph nodes – the set of detected objects, we proceed to detecting their relations, mainly following the approach of Yang et al. [86]: First, we create a directed edge for each pair of objects that are in close enough proximity – as long as the relative distance between the objects in both axes is smaller than 15% of the image dimensions (which covers over 94% of the ground truth edges, and allows us to sparsify the graph, reducing the computation load). Once we set the graph structure, we process it through a graph attention network [77] to predict the identities of each relation, resulting in a probability distribution $P_{L+1}$ over the relation type of each edge.

## 6.5 Implementation and training details

We train the model using the Adam optimization method [47], with a learning rate of $10^{-4}$ and a batch size of 64. We use gradient clipping, and employ early stopping based on the validation accuracy, leading to a training process of approximately 15 epochs, equivalent to roughly 30 hours on a single Maxwell Titan X GPU. Both hidden states and word vectors have a dimension size of 300, the latter being initialized using GloVe [69]. During the training, we maintain exponential moving averages of the model weights, with a decay rate of 0.999, and use them at test time instead of the raw weights. Finally, we use ELU as non-linearity and dropout of 0.15 across the network: in the initial processing of images and questions, for the state and edge representations, and in the answer classifier. All hyperparameters were tuned manually (from the following ranges: learning rate $[5 \cdot 10^{-5}, 10^{-4}, 5 \cdot 10^{-4}, 10^{-3}]$, batch size $[32, 64, 128]$, dropout $[0.08, 0.1, 0.12, 0.15, 0.18, 0.2]$ and hidden and word dimensions $[50, 100, 200, 300]$ to comply with GloVe provided sizes).

We have preprocessed all the questions by removing punctuation and keeping the top 5000 most common words, and use the standard training/test splits provided by the original datasets we have explored [41, 3]: For GQA, we use the more common "balanced" version that has been designed to reduce biases within the answer distribution (similar in motivation to the VQA2 dataset [27]), and includes 1.7M questions split into 70%/10%/10% for training, validation and test sets respectively. For VQA-CP, we use version v2 which consists of 438k/220k questions for training/test respectively. Finally, for the new generalization split, we have downsampled the data to have a ratio of 80%/20% for training/test over approximately 800k questions overall. All results reported are for a single-model settings, except the ensemble scores for GQA that compute majority vote over 10 models, and the ablation studies that are performed over 5 runs for each ablated version. To measure the confidence of the results, we have performed additional 5 runs of our best-performing model over both the GQA Test-Dev and VQA-CP, getting standard deviations of 0.22 and 0.31 respectively.

In terms of time complexity, the NSM is linear both in the number of question words $P$, and the size of the graph ($S \leq 50$ states and $E \approx O(S)$ due to the proximity-based pruning we perform while constructing the graph), multiplied by a constant $N = 8$ of reasoning steps $O(N(V + E + P))$. Similarly, the space complexity of our model is $O(V + E + P)$ since we do not have to store separated values for each computation step, but rather keep only the most recent ones.