[Reviews · NeurIPS 2019]

Reviewer 1



1. Recent parallel work by Haurilet et al. "It’s not about the Journey; It’s about the Destination: Following Soft Paths under Question-Guidance for Visual Reasoning", CVPR 2019 is quite related as it also builds reasoning on scene graphs. However, this was published after NIPS deadline and deals mostly with synthetic images. 2. The supp. material provides ablation studies. In particular, it is good to see a 4% point difference between using region features vs. region concept labels -- highlighting that the symbolic approach is indeed useful. A few additional ablation experiments: (a) Analyze the impact of errors in scene graph prediction. This seems like an important backbone especially as concept labels are being used rather than features. Is there any chance for recovery if the visual prediction makes mistakes? (b) Bias in image question answering is well known where answering works well even without looking at the image. While VQA-CP does limit this to some extent, the proposed method uses a concatenation of the question and the "answer vector" m. What would be the performance without this concatenation? With multiple steps of reasoning, one could hope that this may not be as required as other models. 3. Overall the paper is well written and clear to understand. Code will be released. It might be nice to include one of the qualitative results from the supp. material as it highlights how the proposed approach works. -------------------- Post-rebuttal: I'm happy to recommend accepting this paper. The rebuttal clarifies some of the additional questions raised by all reviewers.

Reviewer 2



I think this is a strong and interesting submission. The presented model, named "Neural State Machine" deviates from the existing approaches to visual question answering by doing 'a sequence of computations' that resembles 'a sequence of reasoning steps'. However, definitely, there are some resembles with the already existing approaches towards visual question answering. E.g., there are approaches that were using outputs of classifier as a knowledge representation, semantic parsers as computational and compositional methods to derive an answer, and use classifier uncertainty to represent concepts (e.g., "A Multi-World Approach to Question Answering about Real-World Scenes based on Uncertain Input"). There are also similarities to graph neural networks in the terms of compositionality, and computability (message passing). However, still the method seems to diverge significantly from these to consider it as a novel. Moreover, the results on GQA are quite strong. The following paper "Language-Conditioned Graph Networks for Relational Reasoning" comes to my mind in terms of the graphnet-equivalent of the Neural State Machine, but it performs significantly worse than the latter. However, this could be a result of different vision representation. It is worth mentioning that GQA is semi-synthetic (questions are synthetic), and hence there is a possibility to 'game' the dataset. Therefore it is nice the authors also provide strong results on the VQA-CP dataset, proving their point. In overall, I think this is interesting submission, with reasonable novel model and strong results.

Reviewer 3



*Originality* The presented approach is relatively easy to understand and doesn’t require extra training data. As far as I can tell, the model is relatively simple and is mostly operating over and recomputing probability distributions of discrete elements in the image and tokens in the sentence. It’s not a surprising next step in this area, but this approach is a good step in that direction. One concern is assumptions placed on the image content space by using a dataset like Visual Genome/GQA. Visual Genome uses a fixed ontology of properties and possible property values and (as the paper states in L129) ignores fine-grained statistics of the image (e.g., information about the background, like what color the sky is). Requiring this fixed ontology may work for a dataset like GQA, which is generated from such an ontology, but may be harder to extend to other, more realistic datasets where topics don’t have to be limited to objects included in the gold scene graph. (Of course, the VQA-CP results are SOTA as well. Thus, I would have liked to see more analysis from VQA-CP, where as far as I understand gold scene graphs are not available.) A highly related paper is Gupta and Lewis 2018 (evaluate on CLEVR by creating a differentiable knowledge graph). *Quality* As stated above, my main concern is that this method relies on a scene graph parser that uses a fixed ontology of object types, properties, and relations. It’s also not obvious to me how the state machine could capture wide scopes as instantiated by universal quantifiers, negation, or counting, which are not heavily represented in GQA (Suhr et al. 2019). Evaluating on additional challenging visual reasoning datasets with real language (and new images never before evaluated by scene graph parsers) could measure this model’s ability to handle a wider and noisier variety of language reasoning skills. Some such datasets: CLEVR-Humans (Johnson et al. 2017) and NLVR (Suhr et al. 2017) would both provide gold standard scene graphs but would evaluate other linguistic reasoning problems; NLVR2 (Suhr et al. 2019) would test both (more than VQA; very recent paper). It would have been nice to see some evaluation of the amount of noise allowable in the state machine (coming from the scene graph). I.e., through perturbing the distributions of properties, or even adding and removing objects. Another way of putting it: how many errors are caused because the scene graph prediction is incorrect, and when evaluating on noisier images (i.e., ones which have *never* been seen by a scene graph parser; as far as I understand it all images in the GQA test set have been seen at least a few times during test-set evaluation of existing scene graph parsers, so existing scene graph parsers should do modestly well on them). ---> after reading the rebuttal, sorry I wasn't aware that the test set was a completely new set of images for GQA. I appreciate the inclusion of experiments wrt. scene graph noise -- never mind on the concern about the test set! What’s the time complexity of applying an instruction to the current distribution over nodes? Since the state machine is a fully-connected graph (as I understand), it seems like this will be a very expensive operation as the scene graph gets larger. What’s the size of the splits for the GQA generalization tests (Table 4)? *Clarity* Details on how the model is trained should be included in the main paper. The approach section left me with many questions about annotation and what parameters were being learned (this should be listed somewhere). It would be naturally followed by a training section. Similarly, experimental setup (e.g., which scene graph parser is used) should be included. The output space of the decoder in Section 3.3 should be defined -- is this picking items from the input hidden states (if so, how is this supervised?)? Or is it just applying a recurrent transformation on them somehow and generating a sequence the same length as the input? Some terminology was confusing as the terms were being overloaded: . “alphabet” -- usually refers to single graphemes, not words. . “learning” in Section 3 (i.e., constructing the state machine) is not really learning anything, just performing a bit of inference to get the state machine. . “raw dense features” should be specified as the direct image features (L125) . “tagging” in L166 to me is a misuse of the term because if anything it’s a soft tag. “Alignment” would be a better term. . “normalizing” in Section 3.3 is also misused; “contextualized” would make more sense. Some other notational things: . The numbers on the edges in Figure 1 don’t seem to add anything. . The types of the elements of the sets in L108--112 should be defined. Are these all vectors? . L in L119 is unexpected without having defined the set of L properties first. . L289: “4th” should be “5th” Typo: “modaility” in L129 *Significance* This approach is easy to understand and doesn’t require extra data beyond scene graph annotation during training. It outperforms SOTA for two visual reasoning tasks.

[Author Response · NeurIPS 2019]



Table 1: Error Analysis

| Type | Rate % |
|---|---|
| Ambiguity / Synonymity | 31 |
| Perception | 29 |
| Counting | 12 |
| External Knowledge | 11 |
| Subjective | 9 |
| OCR | 8 |

Table 2: Noise Experiments

| Alteration | Sem | Sem+Vis |
|---|---|---|
| Add Object | 24/25 | 25/25 |
| Remove Object | 18/25 | 21/25 |
| Remove Attribute | 21/25 | 25/25 |
| Remove Relation | 19/25 | 23/25 |

Dear reviewers and area chairs,

First of all, we would like to thank the reviewers for the feedback, questions and comments! We appreciate a lot the time they took to read our paper and write the reviews. In the following, we discuss specific points made by the reviewers:

**Reasoning Skills and Ablations**. To explore reasoning skills that are not covered by GQA, we compared our model with the BottomUp model for counting and negative (*not*) questions selected from VQA2. In both cases, our approach achieves better performance, with 52.14% vs 48.64% for counting, and 54.2% vs 38.1% for negation. Since the NSM computes a sequence of soft attentions over the graph, it can distribute its attention probability to capture multiple nodes at the same time, which we believe could help with counting, negations or universal quantification. Following reviewer 1's suggestion, we have also explored the impact of the final-stage question conditioning on the overall performance, yielding an accuracy of 60.41% for GQA and 44.35% for VQA-CP. Introducing such conditioning allows the model to consider more directly question nuances that may be necessary to address it, and thus potentially provides increased flexibility in handling more varied reasoning types (even though it may indeed also leverage training biases). To get further insight, we plan to perform a qualitative analysis to inspect the model's internal behavior and produced attention maps for questions involving different reasoning skills.

**Noise Robustness and the Test Set.** To evaluate the model's robustness to noise in the predicted scene graphs, we made graph alterations for 100 questions answered correctly by the model, and measured their impact on its performance. We also evaluated a variant of the NSM model where attention for each object is computed both over its semantic representation and its visual features. We can see in table 2 that even when noise is introduced, both model versions still answer most questions correctly, with the new variant being naturally more robust as it relies on additional information. We plan to extend this initial qualitative assessment and explore larger-scale controlled experiments about noise robustness which we will add to the paper. Finally, please note that, as discussed in the official website, the images and scene graphs in the GQA test set have not been made public and do not overlap with Visual Genome, but rather have been collected independently for the GQA task. As such, they have not been used either for training or evaluation of our or other scene graph parsers – namely, our model's performance over the test set is based on graphs predicted over new unused images, providing further evidence for the ability of the model to cope with potentially noisy graphs.

**Additional Datasets and Error Analysis.** We have begun to explore the model in context of other VQA datasets: Currently, we achieve 68.2% for VQA2 and believe that with tuning we may be able to improve that result. Based on the error analysis summarized in table 1, we can see that many of the questions which are counted as errors (31%) are in fact cases where the model's prediction is semantically equivalent to or synonymous with the labeled answer, e.g. *korean air / korean*, or *flushing / to flush*. Other cases include subjective questions (*Is this a happy home? maybe*) or ones that require external knowledge (*Where is this located? London*) or OCR (*What is written on . . .* ), with the rest of the errors arising from imperfect perception (29%) or counting mistakes (12%). We plan to also perform experiments on other datasets such as CLEVR-humans or NLVR$^2$. Finally, following the strong generalization results obtained for the new GQA splits (section 4.2), we would especially be interested to test the model in terms of data efficiency, e.g. over CLEVR or VQA, comparing the training set size needed by our and other models to achieve the same accuracies.

**Graph Generation and Sparsity.** We implement our own scene graph parser following common ideas from several public implementations [55, 58, 10]. In particular, following [55], we prune the graphs to make them sparser and reduce the computational load, keeping only the top 300 most likely edges, based on a combination of factors including: class-based prior [58], objects proximity, and graph parser's confidence. We will add to the paper a more detailed description of our scene graph parser, including both further implementation details as well as the training scheme.

**Misc and Editing.** The decoder (section 3.3) is indeed not supervised directly but rather recurrently computes attention, first over the concepts and then over the updated question word embeddings, to generate a sequence of soft operations. In lines L108–112 all the elements are vectors of the same dimension $d = 300$: the concept embeddings are learned parameters, while the states, edges and instructions are computed throughout the model operation. The sizes of the new generalization splits are: 834k/241k train/test for the content split and 858k/217k for the structure split. Finally, we will extend the Related Work section to discuss the papers mentioned in the reviews, add the reasoning visualization (Supplementary figure 1) to the main paper, and fix the notation and terminology suggestions.

Thank you very much!
– Paper 3180 authors

[Meta-Review · NeurIPS 2019]

After author feedback and reviewer discussion, this paper received three final ratings of 8 (clear accept). Reviewers supported the general aim of combining neural and symbolic approaches, and were convinced by the strong empirical results on both the GQA and VQA-CP datasets, and the additional experiments provided in author feedback. The AC agrees with reviewers that this paper will be of significant interest to the NeurIPS vision-and-language community. However, we still encourage the authors to pay close attention to the detailed feedback provided by reviewers, in order to further polish the manuscript.